# Update on Omega-3 Polyunsaturated Fatty Acids on Cardiovascular Health

**DOI:** 10.3390/nu14235146

**Published:** 2022-12-03

**Authors:** Daniel Rodriguez, Carl J. Lavie, Andrew Elagizi, Richard V. Milani

**Affiliations:** John Ochsner Heart and Vascular Institute, Department of Cardiovascular Diseases, Ochsner Clinical School—The University of Queensland School of Medicine, New Orleans, LA 70121, USA

**Keywords:** omega-3 FA, fatty acids, cardiovascular disease

## Abstract

Twenty percent of deaths in the United States are secondary to cardiovascular diseases (CVD). In patients with hyperlipidemia and hypertriglyceridemia, studies have shown high atherosclerotic CVD (ASCVD) event rates despite the use of statins. Given the association of high triglyceride (TG) levels with elevated cholesterol and low levels of high-density lipoprotein cholesterol, the American Heart Association (AHA)/American College of Cardiology (ACC) cholesterol guidelines recommend using elevated TGs as a “risk-enhancing factor” for ASCVD and using omega 3 fatty acids (Ω3FAs) for patients with persistently elevated severe hypertriglyceridemia. Ω3FA, or fish oils (FOs), have been shown to reduce very high TG levels, hospitalizations, and CVD mortality in randomized controlled trials (RCTs). We have published the largest meta-analysis to date demonstrating significant effects on several CVD outcomes, especially fatal myocardial infarctions (MIs) and total MIs. Despite the most intensive research on Ω3FAs on CVD, their benefits have been demonstrated to cluster across multiple systems and pathologies, including autoimmune diseases, infectious diseases, chronic kidney disease, central nervous system diseases, and, most recently, the COVID-19 pandemic. A review and summary of the controversies surrounding Ω3FAs, some of the latest evidence-based findings, and the current and most updated recommendations on Ω3FAs are presented in this paper.

## 1. Introduction

Twenty percent of Americans (US) die from cardiovascular (CV) diseases (CVDs) [1]. Five decades after the landmark epidemiological study on the Inuit population of Greenland by Bang et al. [2], the first to show a connection between fish oil (FO) and CV health, CVD continues to be the number one cause of death in the US and the world [3]. CVD generates immense health and economic burdens globally and in the US, as the estimated direct costs have increased continuously from USD 103.5 billion in 1997 to USD 226.2 billion in 2018 [1].

High consumption of polyunsaturated fatty acids (PU FAs), specifically omega-3 FAs (Ω3FAs) such as eicosapentaenoic acid (EPA) and docosahexaenoic acid (DHA), results in low plasma cholesterol levels and minimal coronary heart disease (CHD) [4,5]. Furthermore, as elevated triglycerides (TGs) appear to be a causal factor for atherosclerotic CVD (ASCVD) and possibly for premature all-cause mortality, more so when they are associated with genetic variants, PUFAs can reduce TG levels by decreasing lipoproteins with high amounts of TGs, such as very-low-density lipoproteins, intermediate-density lipoproteins, chylomicrons, and their remnants [6,7].

Numerous studies have investigated Ω3FAs’ benefits on multiple bodily systems, in addition to the CV system. More substantial evidence supports its effects on decreasing plasma cholesterol and lowering the risk of CHD mortality [8], ASCVD, carcinogenesis, central nervous system diseases such as dementia, immune system-related disorders, including rheumatoid arthritis and psoriasis, and possibly having an added benefit in the defense against infections [9,10,11,12,13,14,15]. This article will discuss some of the fundamentals of Ω3FAs, the controversies, and the current recommendations for using Ω3FAs in routine clinical and CV practices.

## 2. Dietary Sources and Guidelines

PUFAs are generally classified into two broad groups: n-3 and n-6. PUFAs with double bonds starting at position six from the methyl end are considered Ω6 series, while those starting at position three are regarded as Ω3 series. Inflammatory mediators are produced by n-6 PUFAs, and n-3 PUFAs form neutral or anti-inflammatory signaling molecules. Arachidonic acid lies within the phospholipids that are present in cell membranes, and it is this Ω6 PUFA that plays an essential role in producing eicosanoids in the body. Although PUFAs are considered a family, their potency varies based on whether their origin is plant- or marine-based. The three major and more studied Ω3 PUFAs are alpha-linolenic acid (ALA), EPA, and DHA.

ALA (ALA, 18:3n-3) is an 18-carbon atom FA with three double bonds, found primarily in plants, including flaxseed, soybean, canola oil, chia seeds, walnuts, and flax [16,17,18]. It is considered an essential FA as humans cannot synthesize it, and it is an important source of very long-chain PUFAs such as EPA and DHA (Figure 1). Apart from being able to be synthesized from plants, EPA and DHA are found in fish, more so in FOs coming from herring, mackerel, trout, salmon, and sardines. To assess the amount of EPA and DHA in the body, red blood cell (RBC) FA composition is used to reflect the cellular membranes throughout the body. A convenient method for relating long-chain Ω3 PUFA levels with CHD risk was proposed by linking the sum of Ω3 PUFAs (EPA and DHA) in RBCs to the Ω3 index (O3I) [19,20,21], finding an inverse correlation of the O3I with CHD. Accordingly, the American Heart Association (AHA) recommends eating 3 ounces of cooked fish (particularly fatty fish) two times per week [22]. Despite recommendations and the known benefits of Ω3FA, in the periods 2005–2006 and 2015–2016, the National Health and Nutrition Examination Survey reported a 6.8% decrease in the number of Americans older than twenty years old who consumed seafood more than once a week [23] (Figure 2).

The characteristics of patients with lower seafood consumption were examined by Love et al. [24], finding that individuals with lower incomes consumed less seafood (120.2 g/week) than individuals from high-income groups (141.8 g/week), with even lower consumption of seafood containing long-chain n-3 PUFAs (lower income: 21.3 g/week. vs. higher income 46.8 g/week) [24]. These results were later corroborated after adjusting for age and sex as potential confounders, again finding that people with lower income consumed 18% less seafood than the people with higher income (*p* = 0.03), with an added lower intake of nuts, seeds (*p* < 0.001), soy (*p* < 0.001), and all protein foods (*p* < 0.001). There were several reasons attributed to this difference; one of them was the price difference between fresh seafood with high n–3 PUFAs and those with low n–3 PUFAs. Fresh seafood with high n–3 PUFAs was 32% more expensive than fresh seafood with low n–3 PUFAs (*p* < 0.002). These findings could explain why prior authors have found an inverted independent link between socioeconomic status and the risk of ASCVD [25,26,27,28,29].

Compliance with diet should be a priority, as healthcare costs associated with increasing 20% adherence to diet are estimated to result in annual cost savings of approximately USD 31.5 billion. Cost savings related to CVD account for half of these savings [1].

## 3. Molecular Mechanisms

Cell membrane phospholipids contain FAs, which play a significant role in various functions, metabolic reactions, and signaling processes. Different levels of PUFAs in cell membranes are required to exert actions and to maintain proper functioning and tissue responsiveness to signaling. These levels depend on a sufficient intake of PUFAs from the diet.

Cellular cholesterol requirements are satisfied by either exogenous cholesterol inflow paths involving multiple lipoprotein receptors or endogenous cholesterol synthesis, which is regulated by the 3-hydroxy-3-methylglutaryl-coenzyme A reductase (HMGR), a rate-limiting enzyme. The catabolism of cholesterol is exerted by the upregulation of the liver’s low-density lipoprotein (LDL) receptor (LDLR) gene and protein expression. A combination of the LDL and LDLR binds to the ligand and internalizes in lysosomes. The ligand uncouples from the receptor due to the lower pH of the lysosomes. In this process, the lipids in the receptor are degraded, and the receptor is then returned to the cell membrane for further lipid binding.

Through a negative feedback mechanism tightly controlled by two proteins, the LDLR gene transcription is regulated by cholesterol availability, the sterol regulatory element-binding protein (SREBP), which requires cleavage to become active, and the SREBP cleavage-activating protein (SCAP), which cleaves it for activation. The SCAP proteins transport SREBPs to the Golgi apparatus when cholesterol levels in cells are low. Once in the Golgi apparatus, they are cleaved in order to activate enzymes involved in the synthesis of lipids. It has been shown that the sterol sensor protein SCAP undergoes conformational changes in response to high cholesterol levels in the cell, allowing it to adhere to the endoplasmic reticulum protein Insig-1, forming a ternary complex known as SREBP/SCAP/Insig-1, trapping it in the endoplasmic reticulum, stopping cholesterol synthesis and uptake (via the LDLR) and maintaining cell cholesterol homeostasis. As demonstrated through posttranslational downregulation of the LDLR, PCSK-9 plays a significant role in cholesterol metabolism by binding specifically to the LDLR on the cell membrane in order to form a PCSK9/LDLR complex, which then prevents the recycling of the LDLR by redirecting it to the lysosome, where it will be degraded. The disruption of this pathway can result in the accumulation of cholesterol and the formation of foam cells in organs.

ALA promotes cholesterol conversion into bile acids by the cholesterol 7α-hydroxylase (CYP7). As a result of removing hepatic cholesterol from the circulation by the synthesis of bile acids, ALA promotes SREBP activation (via SCAP activity), upregulating LDLR expression and favoring the clearance of LDL cholesterol from the body. Ω3FAs such as EPA and DHA reduce TG levels by direct inhibiting liver diacylglycerol acetyl-transferase, which catalyzes the formation of TG from diacylglycerol, essential for TG intestinal absorption and fatty acyl-CoA, and the inhibition of the phosphatidic acid phosphohydrolase required for triacylglycerol (TAG) synthesis from glycerol 3-phosphate [30]. Other molecular effects are exerted by inhibiting the acyl-CoA:1,2-diacylglycerol acyltransferase, increasing mitochondrial and peroxisomal-beta-oxidation in the liver, decreasing lipogenesis, and increasing plasma lipoprotein lipase activity. EPA’s ability to stabilize cell membranes, along with its ability to lower cholesterol, may contribute to the 30% and 40% reductions seen in deaths from CVD and sudden cardiac death (SCD) and the 56% reduction in cardiac arrest [31].

Ω3FAs lower TG-rich lipoproteins and increase anti-aggregatory and vasodilatory prostanoids such as prostacyclin, combating thrombosis and vasospasm. It can incorporate into the mitochondria and plasma membranes, stabilizing them and preventing them from oxidation, which is believed to have a role in preventing arrhythmias. Additionally, Ω3FAs’ are precursors to the synthesis of specialized mediators capable of combating inflammation and have been demonstrated to decrease proinflammatory cytokines such as interleukine-6 and tumor necrosis factor-α and inhibit the activation of the ikappaB kinase and nuclear factor-κB, as well as several other transcription factors that inhibit reactive oxygen species [32,33]. These anti-inflammatory properties are believed to interfere less with self-defense than direct anti-inflammatory treatments. A combination of these mechanisms is believed to contribute to the CVD protection associated with Ω3FA consumption and the added benefit on multiple other systems and pathologies [34].

## 4. Controversies Surrounding Ω3

It has been debated since the Inuits study whether Ω3FA consumption is solely responsible for the CV benefits observed in this population, whether fish intake is beneficial on its own, or whether an overall healthier diet resulting from a higher fish intake is beneficial in lowering CVD risk. The Diet and Reinfarction Trial (DART) [35] was the first RCT to show a reduction in mortality during the two years after myocardial infarction (MI) among men who were advised to eat about 300 g of FO per week or who took an equivalent amount of n-3 fatty acids in the form of FO supplements. Later, these findings were confirmed by the GISSI– Prevenzione trial [36], the Lyon Diet Heart Study [37], and various cohort studies.

The OMEGA trial tested the effects of adding Ω3-acid ethyl esters-90 (1 g/d for one year) to current guidelines. The primary endpoint was SCD in survivors of acute MI. Secondary endpoints were non-fatal clinical CVD events and total mortality. The patients were followed up for 365 days; in this study, investigators found no difference between omega and control groups in the rates of SCD (1.5% and 1.5%; *p* = 0.84), total mortality (4.6% and 3.7%; *p* = 0.18), major adverse cerebrovascular and CVD events (10.4% and 8.8%; *p* = 0.1), and revascularization in survivors (27.6% and 29.1%; *p* = 0.34); however, a significant limitation of this trial was a lack of statistical power and a reduced rate of SCD, total mortality, and major adverse CVD events (MACE) after one year of follow-up. In 2019, the strength of Ω3FAs was again demonstrated in the RCT REDUCE-IT trial, demonstrating the added benefit of consuming 4 g/day of icosapent ethyl (IPE), divided into twice-daily doses (2 g two times per day), to reduce ischemic events and CVD death [38].

While individual trial results are inconsistent, the results of pooled RCTs suggest a cardioprotective effect of Ω3FAs. The possible explanation for this heterogeneity includes differences in dosages between groups, the differences between study follow-ups, sample sizes, and lower event rates [39]. Further controversies regarding the effect of DHA on preventing CVD were drawn after some studies found rising LDL-C among patients given DHA. This controversy was later clarified in clinical trials, where using adequate doses of 4 g/day failed to demonstrate any increase in LDL cholesterol [40,41,42,43,44].

Ω3FAs have also been discussed for their arrhythmogenic effects. Initial studies in animal labs demonstrated that supplementation with DHA, but not EPA, reduced arrhythmogenic structural changes to the atria resulting from simultaneous atrial and ventricular pacing [45]. However, four studies providing a combination of EPA–DHA to assess the risk of AF suggested, but did not prove, that the risk of AF with Ω3FA intake may be dose-related. Doses of 1.8 g/d had an increase in risk (hazard ratio (HR), 1.84) of atrial fibrillation (however, not achieving statistical significance), and doses of 4.0 g/d almost doubled the risk of this arrhythmia [5,46,47,48]. These findings were later contradicted by The MESA study (Multiethnic Study of Atherosclerosis), which examined the relationship between Ω3FAs (expressed as a percentage mass of total fatty acids) and the risk of major bleeding events and AF [49]. This study included a population free of CVD, finding that higher DHA levels were associated with fewer incidents of AF (HR, 0.80; CI, 0.65–0.98; *p* = 0.03) and that higher EPA and DHA were associated with significantly fewer hospitalizations for bleeding events (EPA (HR, 0.75 CI, 0.60–0.94, *p* = 0.01; EPA + DHA (HR, 0.84; CI, 0.73–0.98; *p* = 0.03)) [5,50].

## 5. Index

A diet low in Ω3FAs is associated with increased mortality and CHD risk [5,38,51,52], although, as concentrations of Ω3 vary among varieties of food and among types of fish and vegetables, measuring the amount of Ω3 consumed by a person is not a reliable indicator of the level of Ω3 in the organism. In addition, various factors can affect the blood levels of Ω3FAs after eating an Ω3-rich meal, such as the variability of the uptake of ingested EPA and DHA and the difference in bioavailability between individuals (EPA and DHA in pregnant or obese women compared to EPA and DHA in lean women). Once absorbed, Ω3FAs become part of the cell membranes and have the ability to affect several of their properties, including modulating the activity of membrane-bound enzymes and cell signaling pathways [53,54]. Therefore, to better assess the amount of Ω3FAs in the human body, researchers have proposed using the O3I. This term was first introduced in 2004 by William S Harris and Clemens Von Schacky [55], who described the O3I as the percentage of the total red blood cell membrane formed by EPA and DHA. Its levels were demonstrated to have a direct correlation with the risk of death from CHD, where an O3I value of 8% decreased the risk of CHD mortality and a percentage less than 4% increased this risk [55].

## 6. Recent Data

In 2018, a meta-analysis of ten trials involving 77,917 participants found that consuming marine-derived Ω3FAs over 4.4 years did not lead to a significant decrease in CHD. According to the study, Ω3FA supplements did not prevent fatal MIs and strokes in people with a high CVD risk [56]. Later in 2019, the New England Journal of Medicine published the results of the REDUCE-IT trial, a multicenter, double-blind, randomized, placebo-controlled trial (RCT) that included 8179 patients with risk factors for CVD or established CVD who were treated with statin therapy. The included cohort had fasting TG levels of 135 to 499 mg/dL and LDL cholesterol levels of 41 to 100 mg/dL (1.06 to 2.59 mmol per liter). They were followed over a period of 4.9 years [5], where one group of patients received 2 g of IPE two times per day and the other group received placebos. The primary composite endpoint of CVD death, non-fatal MI, non-fatal stroke, coronary revascularization, or unstable angina (UA) occurred in 17.2% vs. 22.0% of patients in the IPE group vs. the placebo group (HR, 0.75; 95% confidence interval (CI), 0.68 to 0.83; *p* < 0.001). The secondary outcome, which was a composite of CVD death, non-fatal MI, or non-fatal stroke, happened in 11.2% of subjects taking IPE and in 14.8% of those taking placebos (HR, 0.74; 95% CI, 0.65 to 0.83; *p* < 0.001), demonstrating the added benefit of 2 g of IPE twice daily on reducing the risk of ischemic events and CVD death (Figure 3).

Following the REDUCE-IT trial and due to the controversial results between RCTs and the different dosages used, a meta-analysis of 13 RCTs with 127,477 participants was conducted and published in October 2019 [51]. In order to determine the correlation between the dose of marine Ω3 supplements and the risk of the specific prespecified outcome, a meta-regression analysis was carried out. Differences between studies were evident, not only for a variation in the dose of marine omega-3 supplementation (ranging from 376 to 4000 mg/dL) but also variations in the proportions of EPA and DHA. Even when excluding the REDUCE-IT trial, marine Ω3 supplementation was associated with a significantly lower risk of MI (rate ratio (RR) (95% CI): 0.92 (0.86, 0.99); *p* = 0.020), CHD death (RR (95% CI): 0.92 (0.86, 0.98]); *p* = 0.014), total CHD (RR (95% CI): 0.95 (0.91, 0.99); *p* = 0.008), CVD death (RR (95% CI): 0.93 (0.88, 0.99]); *p* = 0.013), and total CVD (RR (95% CI): 0.97 (0.94, 0.99); *p* = 0.015). After including the REDUCE-IT trial, the findings were strengthened, finding a linear dose–response relationship for total CVD and major vascular events, where every 1000 mg/d marine omega-3 supplementation corresponded to 9% (95% CI: 2%, 15%; *p* = 0.012; *p* for heterogeneity 0.218) and 7% (95% CI: 0%, 13%; *p* = 0.041; *p* for heterogeneity 0.068) lower risks of MI and total CHD, respectively. The conclusion was similar to the REDUCE-IT trial, where marine Ω3 supplements lowered the risk for MI, mortality, and the total number of CHDs; additionally, mortality and the total number of CVDs were lowered even when the REDUCE-IT trial was excluded from the metanalysis. Similar studies have been published over the last few years, supporting that Ω3FAs reduce CVD mortality (RR, 0.93 (0.88–0.98); *p* = 0.01), non-fatal MI (RR, 0.87 (0.81–0.93); *p* = 0.0001), number of CHDs (RR, 0.91 (0.87–0.96); *p* = 0.0002), major adverse CVD events (RR, 0.95 (0.92–0.98); *p* = 0.002), and revascularization (RR, 0.91 (0.87–0.95); *p* = 0.0001) [58].

On April 2021, a meta-analysis was conducted, studying baseline plasma Ω3FA composition (ALA, EPA, docosapentaenoic acid, and DHA) assessed through gas chromatography. The study included patients with CVD death (*n* = 203), MI (*n* = 325), ventricular tachycardia (*n* = 271), and AF (*n* = 161) and 1612 patients from MERLIN-TIMI 36 who had no events. The latter patients were used as controls. This study concluded that among patients taking Ω3FAs, higher long-chain Ω3-PUFAs were inversely associated with lower odds of SCD (adj OR, 0.91; 95% CI, 0.67–1.25), independent of traditional CVD risk factors and lipids [59]. Similarly, among patients given IPE, there was a 40% reduction in the risk of repeated coronary revascularization (17.1% vs. 27.6%; HR, 0.60; 95% CI, 0.51–0.70; *p* < 0.001). The rates of the primary composite endpoint of CVD death, non-fatal MI, non-fatal stroke, coronary revascularization, or UA requiring hospitalization were 20.8% among patients administered IPE vs. 29.4% among patients given placebos (HR, 0.66; 95% CI, 0.58–0.76; *p* < 0.001), representing a 34% relative risk reduction, an 8.5% absolute risk reduction, with and a number needed to treat (NNT) of 12 patients to prevent 1 MACE event over a median of 4.8 years. There was also a 34% reduction in the key secondary composite endpoint of CVD death, non-fatal MI, or non-fatal stroke (HR, 0.66; 95% CI, 0.56–0.79; *p* < 0.001; NNT4.8 years = 19), with a 39% reduction in total events (rate ratio, 0.61; 95% CI, 0.52–0.72; *p* < 0.001).

In 2022, Zheng Gao et al. [60] conducted a meta-analysis that included 21 RCTs and one observational study to determine the effect of Ω3 PUFAs on coronary atherosclerosis. The total population of the study was 2277 participants. Participants who consumed Ω3 PUFAs had a decrease in the progression of CAD by reducing the volume of atherosclerotic plaques (SMD, −0.18; 95% CI, −0.31 to −0.05) as well as keeping the narrowest coronary arteries from further decreasing their diameter (SMD, 0.29; 95% CI, 0.05–0.53). Similarly, in May 2022, the JACC published a post hoc analysis of the REDUCE-IT trial in patients with prior MIs. This analysis examined the benefit of IPE vs. placebo on reducing ischemic events [31], demonstrating a decrease from 26.1% to 20.2% (HR, 0.74, 95% CI, 0.65–0.85; *p* = 0.00001) in the composite of CVD death, MI, stroke, coronary revascularization, and hospitalization for UA. The secondary endpoint of CVD death, MI, or stroke was reduced from 18.0% to 13.3% (HR, 0.71, 95% CI, 0.61–0.84; *p* = 0.00006). Furthermore, relative risk (RR) reduction was calculated for each specific outcome, finding a significant RR reduction of 35% in total ischemic events, a 34% RR reduction in MI, a 30% RR reduction in CVD death, and a 20% lower rate of all-cause mortality. In patients with prior MI taking Ω3FAs, relative and absolute risk reductions were overall decreased for ischemic events, including CVD death.

## 7. Importance of Dosage

It is a known fact that a significant percentage of substances absorbed do not reach the systemic circulation or tissues following their physiological intent. From a pharmacokinetic and dietary planning perspective, this is a crucial difference. Fortunately, blood plasma, serum, blood cells, and lymph can be measured for Ω3FA concentrations, where blood cell fatty acid concentrations indicate long-term bioavailability. In contrast, blood plasma fatty acid concentrations reflect the short- to medium-term supply of fatty acids in the diet [61,62]. However, finding the threshold to which there are still benefits from taking nutritional supplements, vitamins, and medications is always challenging.

Numerous studies have been published examining Ω3FA dosages. The Effects of EPA on Major CHD Events in Hypercholesterolemic Patients (JELIS) trial was a randomized, open-label, blinded endpoint analysis conducted in Japan between 1996 and 1999 [50]. Patients were randomly assigned to a daily dose of 1800 mg of EPA with a statin (*n* = 9326) or statin only (*n* = 9319). After five years of follow-up, in patients with a history of CHD who were given EPA treatment, major CHD events were reduced by 19% (*p* = 0.04). Two decades later, in 2019, two RCTs were published in the NEJM: the Vitamin D and Ω3 Trial (VITAL) RCT and the REDUCE-IT trial. The VITAL trial took place at the Brigham Women’s Hospital using a two-by-two factorial design, and it examined whether people with no history of cancer, CVD, or stroke could reduce their risk of developing these illnesses by taking vitamin D supplements at a dose of 2000 IU per day (in the form of vitamin D3—cholecalciferol) and/or marine Ω3FAs (EPA and DHA) at 1 g per day. A total of 25,871 participants, including 5106 black participants, were enrolled. Participants were randomly assigned to one of four groups: daily vitamin D and Ω3; daily vitamin D and Ω3 placebo; daily vitamin D placebo and Ω3; or daily vitamin D placebo and Ω3 placebo. This study did not show that supplementing with n-3 fatty acids reduced the incidence of major CVD events (a composite of MI, stroke, and death from CVD causes) or invasive cancer. However, this trial showed that just 840 mg of combined EPA/DHA significantly reduced MI and major CHD events [63].

In contrast, Deepak L. Bhatt et al., in the REDUCE-IT trial [5], used four times the dose of Ω3FAs (IPE 2 g twice daily) used in the VITAL trial. With a twice-daily dose of 2 g of IPE, patients on the IPE were less likely to develop ischemic events, including CVD death. These observations were independent of the use of statins.

Based on a meta-analysis and meta-regression of 40 interventional trials involving 135,267 participants, Ω3 proved to have dose-dependent significant protective effects against CVD and MI [57]. The meta-regression analysis found that increasing intake by 1 g/day was associated with a statistically significant reduction of 5.8% in the risk of CVD events, with an additional statistically significant risk reduction of 9.0% in MIs for each additional 1 g/day of Ω3. However, since this meta-analysis, two major trials have been published with negative results [47,48]. Therefore, we have updated our meta-analysis with these two trials (now 42 trials with nearly 150,000 participants), and the overall results still demonstrate marked reductions in major CVD events, especially fatal MI and total MI. However, different from the initial meta-analysis, for every increase of 1 g/d of DHA and EPA, there was a 9% reduction in MI but no reduction any longer for CVD events (Figure 4) [52]. As demonstrated in our meta-analysis and the REDUCE-IT trial, for patients with hyperTG, clinicians and patients are advised to consider the potential benefits of Ω3 supplementation at a dose of 2 g of EPA/DHA twice daily (total daily dose, 4 g) to reduce the risk of major ischemic events, including CVD death.

## 8. Chronic Kidney Disease (CKD)

When studying the effects of Ω3FAs on patients with CKD, a retrospective analysis of 2990 participants with CKD used a multivariable Cox proportional hazards model to study the association between dietary Ω3 PUFAs and mortality [64]. The adjusted HRs (95% confidence interval) for all-cause mortality of the diseased people with CKD in the second (0.87–1.30 g/day), third (0.87–1.30 g/day), and fourth (1.93–9.65 g/day) quartiles of dietary Ω3 PUFAs were 0.94 (0.72, 1.23), 0.74 (0.54, 1.02), and 0.67 (0.48, 0.93), respectively, versus those with the lowest quartile of dietary Ω3 PUFA intake (<0.86 g/day) (*p* for trend = 0.011). This study suggests an inverse relationship between dietary Ω3 PUFA intake and all-cause mortality in patients with CKD.

## 9. Heart Failure (HF)

Oxidative stress is a significant cause of heart fibrosis; it is highly implicated in the development of HF and is partly controlled by the nuclear factor erythropoietin 2 related factor 2 (NRF2). EPA, DHA, and specialized pro-resolving lipid mediators (SPMs) such as resolvin D1 (RvD1) can activate NRF2, which could protect the heart from the onset of cardiac fibrosis. Furthermore, the Ω3 protective effects against CHD and MI in primary prevention could translate into averting HF in the long term since CHD and MI are associated with the pathophysiology of HF [52]. Therefore, to assess the benefit of Ω3FAs in HF, Senthil Selvaraj et al. analyzed patients in the REDUCE-IT trial and, by implementing the Cox regression model, estimated the risk of outcomes of participants with and without HF [65]. The primary endpoint comprised CVD death, non-fatal MI, non-fatal stroke, coronary revascularization, or UA. Among 1446 patients with HF, IPE reduced TG (median reduction, 33.5 mg/dL, or 15.4%; *p* < 0.0001) and high-sensitivity C-reactive protein (35.1%; *p* < 0.0001); its effects were similar to patients without HF (*p*-interaction > 0.90). In addition, the treatment effect on the primary endpoint was consistent among patients with and without HF (HR 0.87, 95% CI 0.70–1.08; HR 0.73, 95% CI 0.65–0.81, respectively; *p*-interaction = 0.13).

Among patients with HF, the RCT demonstrated that a long-term administration of 1 g per day n-3 PUFA (850–882 mg of EPA–DHA) compared to a placebo is effective in reducing both all-cause mortality and admissions to the hospital. Patients who took Ω3FAs had an absolute risk reduction of 2.3% (0–4.6) for mortality or admission for CVD reasons and 1.8% for all-cause mortality (95% CI 0.3–3.9). Among patients who took at least 80% of the doses of Ω3 (4994), the rate of all-cause death was 26% (658 of 2512) in the n-3 PUFA group and 29% (725 of 2482) in the placebo group (adjusted HR 0.86, 95.5% CI 0.77–0.95, *p* = 0.004) [66]. A post hoc subgroup analysis revealed that this reduction in mortality and SCD was concentrated in the approximately 2000 patients with reduced left ventricular ejection fraction [67]. Hence the 2022 AHA/ACC guidelines for managing HF have outlined that Ω3FAs might be reasonably used as an adjunctive therapy on patients with HF class II-IV symptoms to reduce mortality and CVD hospitalizations [68]. We hypothesized many years ago that higher doses of Ω3 may be particularly beneficial for advanced HF and cardiac cachexia, and RCTs are still needed in these HF patients [69].

## 10. Brain Effects

Organs, such as the brain and retinas, depend on systemic and dietary sources of Ω3FAs such as DHA and EPA, which represent approximately 20% of FAs in the central nervous system and are essential for the development and functioning of the brain and eyes [70,71,72,73]. Conversion of plant-derived ALA to DHA is minimal in humans; moreover, these fatty acids cannot be synthesized de novo in the brain or retina, and their absorption depends on proteins such as Major Facilitator Superfamily Domain Containing 2A (MFSD2A), which absorbs them in the form of lysophosphatidylcholine [74,75]. Therefore, DHA pre-formed during pregnancy is preferred for brain development and throughout life. Under normal circumstances, these proteins are abundant in the blood–brain barrier and blood–retinal barrier; however, mutations have been identified, resulting in the loss of function of this protein family. In addition, this mutation has been linked to diseases such as autosomal recessive primary microcephaly-15 [76,77].

It is imperative to note that brain function is determined not only by the brain’s structure but also by the perfusion of the brain, which is modulated by vasoactive derivatives of Ω3FAs [34]. The O3I has been demonstrated to correlate with higher brain perfusion, measured by single photon emission computed tomography. Higher brain levels of omega 3 PUFAs and their atherosclerotic plaque stabilizing properties, as well as their anti-venous thrombosis, anti-inflammatory, antiplatelet, and anticoagulant properties, result in fewer risks for embolism and plaque rupture [20,78,79,80]. On the other hand, a lower O3I value is associated with reduced brain volume, impaired cognition, accelerated progression to dementia, and increased risk of ischemic CVA and death [81].

Moreover, as DHA constitutes an essential component of neuronal membranes, changes to these membranes produce modifications to the proteins and receptors embedded within the membrane, hence altering neuronal activity [82]. In patients with major depressive disorder and those with schizophrenia [83], low consumption of Ω3FAs has been identified [84]. Furthermore, when Ω3FA is given as add-on therapy, double-blind, placebo-controlled trials have found an added therapeutic benefit, demonstrated by improvements in the Positive and Negative Syndrome Scale (PANSS) [83,85,86].

## 11. COVID and Future Pandemics

In response to the coronavirus disease 2019 (COVID-19) pandemic, more research has been done examining the morbidity and mortality associated with severe acute respiratory syndrome coronavirus 2 (SARS-CoV-2) infection. Given the profound public health concerns related to COVID-19, years of studies were accelerated in a short time. It is now known that COVID-19 exerts some of its effects by creating a hyperactive immune response characterized by the release of interferons, interleukins, tumor necrosis factors, chemokines, and several other mediators [87]. This cytokine storm is a prominent component of many common diseases, such as arthritis and periodontal disease, as well as inflammatory bowel disease, CVD, the neurodegenerative diseases Alzheimer’s and Parkinson’s, asthma, cancer, metabolic syndromes (e.g., obesity), diabetes, and autoimmune diseases [88].

Ω3FA has been demonstrated to play a role in this cytokine storm by being a substrate of plasma and synovial fluid specialized pro-resolving mediators (SPMs) such as the recombinant bivalent fusion protein (RvEs), protectin D1 or neuroprotectin D1 (PD1), protectin DX (PDX), and the pro-resolving lipid mediator maresin 1 (MaR1), which reduce circulating inflammatory cytokines, thereby reducing inflammatory-mediated diseases such as rheumatoid arthritis (RA) disease activity [88,89]. This was later demonstrated by an inverse correlation between the O3I and disease activity [90].

Asher et al. performed a study on 100 patients to test the hypothesis that the O3I was inversely associated with the risk of death from COVID-19. During admission, the O3I was analyzed; after adjusting for age and sex, the odds ratio for death in patients with an O3I in Q4 vs. Q1–3 was 0.25, suggesting that a relationship exists. Still, the study was underpowered and did not reach statistical significance. However, as indicated by the trend, more well-powered studies are needed to demonstrate the benefit of Ω3 in COVID-19 [91]. As demonstrated in multiple inflammatory diseases and with the observed mortality benefit trend in COVID-19 patients, Ω3FA as an anti-inflammatory mediator could serve as a prophylaxis and co-treatment in possible future pandemics.

## 12. Conclusions

Although studies surrounding Ω3FAs are still controversial, over recent years, multiple studies have demonstrated the benefits of this PUFA on inflammation and TG reduction, which are key aspects of the pathophysiology of CVD. RCT data show mortality and hospitalization benefits among those patients with CHD, HF, and other non-CV pathologies where inflammation plays a significant role. As recommended by AHA/ACC HF guidelines in 2022, to reduce mortality and CVD hospitalizations among patients with HF class II-IV symptoms, Ω3FAs may be suitable as adjunctive therapy. Clearly, some RCTs, especially REDUCE-IT with pure EPA, and our major meta-analysis of 42 studies in nearly 150,000 participants demonstrate reductions in major CVD outcomes with combined EPA/DHA, especially for fatal MI and total MI, but also for total and fatal CHD events. To reduce the risk of CVD mortality, we recommend consuming 2 g/d of Ω3 EPA/DHA. Among patients with ASCVD and hyperTG, we recommend a dose of 2 g of EPA/DHA two times per day (for a total of 4 g daily) to reduce the risk of major ischemic events, including CVD mortality. We acknowledge that the management of CVD is evolving, thus further RCTs are necessary to gain a deeper understanding of the use of Ω3FAs for the management of dyslipidemia and CVD in the hope of further reducing morbidity and mortality as well as the burden of healthcare associated with CVD in the US and worldwide. 

## Figures and Tables

**Figure 1 nutrients-14-05146-f001:**
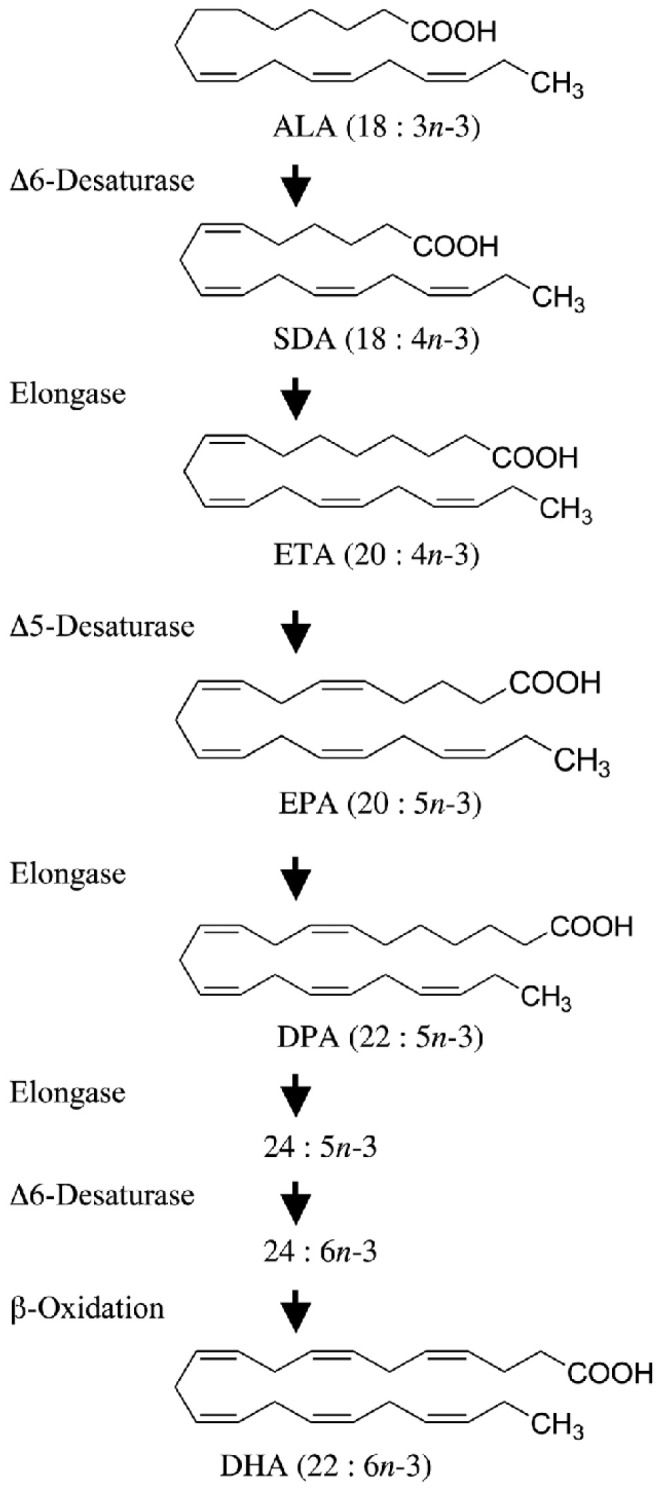
Alpha-linolenic acid (ALA) as a substrate of eicosapentaenoic acid (EPA) and docosahexaenoic acid (DHA).

**Figure 2 nutrients-14-05146-f002:**
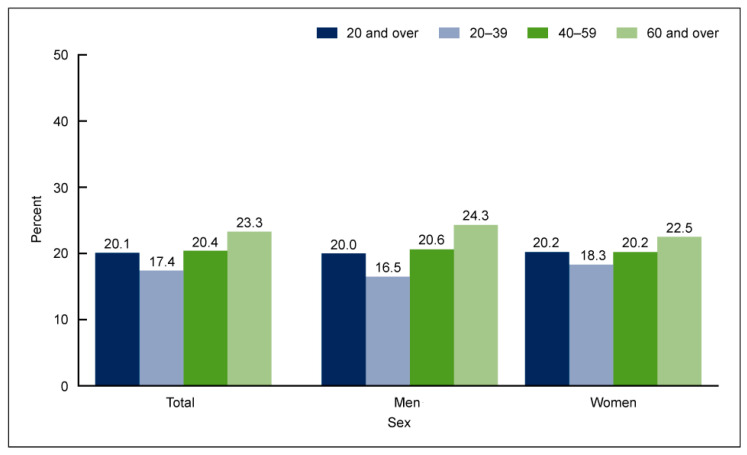
Percentage of adults aged 20 and over consuming seafood at least two times per week, by age and sex, in the United States in 2013–2016 [23].

**Figure 3 nutrients-14-05146-f003:**
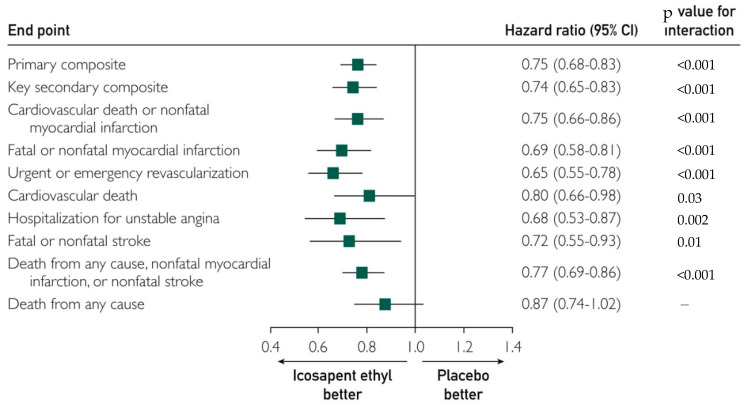
REDUCE-IT trial prespecified endpoints. The rates of all endpoints, except death from any cause, were significantly lower in icosapent ethyl (eicosapentaenoic acid ethyl esters). Data were adapted from NEJM [5] and reproduced with Mayo Clinic Proc [57] permission.

**Figure 4 nutrients-14-05146-f004:**
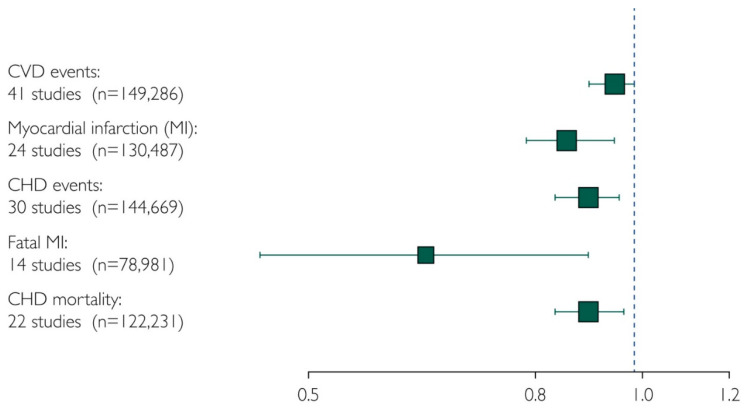
Pooled results from a meta-analysis. Pooled estimate of relative risk and 95% confidence interval, number of studies, and the combined number of participants. CHD, coronary heart disease; CVD, cardiovascular disease; MI, myocardial infarction [52].

## Data Availability

Not applicable.

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
