# Peer review of "Update on Omega-3 Polyunsaturated Fatty Acids on Cardiovascular Health"

_nutrients, 2022, doi:10.3390/nu14235146_

Round 1
Reviewer 1 Report
This review on omega-3 polyunsaturated FAs and their functional aspects regarding human cardiovascular health issues is very well written and the body text is relevantly organized to clear chapters. Importantly the references come up to this year 2022. Also some controversial results are included and discussed. It is rather evident that omega-3 FAs have multitude of targets and the effects are protective/beneficial especially in cases of ATCVD. This reviewer has only a few issues that need further discussions.
Comments
1. What are the consequences of omega-3 FA incorporation in plasma membranes regarding the functionality of certain important receptors involved in lipid metabolism, i.e. SR-BI, ABCA1, ABCG1, LRP-1?
2. Does feeding of omega-3 FAs affect brown adipose tissue (BAT) function or affect WAT "browning"?
3. Reportedly LPL has substrate specificity for short-chain fatty acids. The authors indicate here that EPA/DHA as long-chain unsaturated FAs enhance post-heparin LPL activity in plasma. This is somewhat discrepant data based on earlier reports. What is the enhancing mechanism of EPA/DHA here?
4. What kind of effects EPA/DHA have on arterial endothelial layer? Can we state that these FAs are endothelotherapeutic?
5. What is the effect of these FAs on macrophage function especially cholesterol efflux process via HDL functioning as cholesterol acceptor?
6. Scientific society at least previously was interested in omega-6/omega-3 FA ratio and what might be optimal. What is the state-of-art in these discussions?
7. Do these omega-3 FAs affect beneficially the glymphatic system, i.e. waste clearance pathway in the brain, dedicated to drain away soluble waste proteins and metabolic products in case of Parkinson and Alzheimer patients? What is the added value to measure EPA/DHA in cerebrospinal fluid?
Author Response
- What are the consequences of omega-3 FA incorporation in plasma membranes regarding the functionality of certain important receptors involved in lipid metabolism, i.e. SR-BI, ABCA1, ABCG1, LRP-1? Thanks for your comments. Although we would love to include all aspects concerning omega 3, we have selected a handful of references and information to support what we believe is key information to be delivered. We would consider addressing your question in other papers.
2. Does feeding of omega-3 FAs affect brown adipose tissue (BAT) function or affect WAT "browning"?
Thanks for your comments. Although we would love to include all aspects concerning omega 3, we have selected a handful of references and information to support what we believe is key information to be delivered. We would consider addressing your question in further papers.
3. Reportedly LPL has substrate specificity for short-chain fatty acids. The authors indicate here that EPA/DHA as long-chain unsaturated FAs enhance post-heparin LPL activity in plasma. This is somewhat discrepant data based on earlier reports. What is the enhancing mechanism of EPA/DHA here?
We have made no comments on heparin.
4. What kind of effects EPA/DHA have on arterial endothelial layer? Can we state that these FAs are endothelotherapeutic?
Thanks for your comments. please refer to line 767 of the new uploaded version of the manuscript. “Ω3FA lower TG-rich lipoproteins and increases anti-aggregatory and vasodilatory prostanoids, such as prostacyclin, combating thrombosis and vasospasm.” Although we would love to include all aspects concerning omega 3, we have selected a handful of references and information to support what we believe is critical information to be delivered.
5. What is the effect of these FAs on macrophage function, especially cholesterol efflux process via HDL functioning as a cholesterol acceptor?
Thanks for your comments. Although we would love to include all aspects concerning omega 3, we have selected a handful of references and information to support what we believe is critical information to be delivered. We have addressed some of the anti-inflammatory aspects and some pathophysiology of the benefits of omega 3 FA in CVD.
6. Scientific society at least previously was interested in omega-6/omega-3 FA ratio and what might be optimal. What is the state-of-art in these discussions?
Thanks for your comments. Although we would love to include all aspects concerning omega 3, we have selected a handful of references and information to support the CVD of omega 3 FA and what we believe is critical information to be delivered. We understand your interest in this topic; however after looking for more information regarding this question, over the last ten years, there has been no RCT on CVD studying the omega-6/omega-3 fatty acids (FA) relationship. The papers are limited to the following:
Bhatt, R. S., Sahoo, A., Karim, S. A., & Agrawal, A. R. (2016). Effects of calcium soap of rice bran oil fatty acids supplementation alone and with DL-alpha-tocopherol acetate in lamb diets on performance, digestibility, ruminal parameters and meat quality. J Anim Physiol Anim Nutr (Berl), 100(3), 578-589. doi:10.1111/jpn.12370
Eastwood, L., Leterme, P., & Beaulieu, A. D. (2014). Changing the omega-6 to omega-3 fatty acid ratio in sow diets alters serum, colostrum, and milk fatty acid profiles, but has minimal impact on reproductive performance. J Anim Sci, 92(12), 5567-5582. doi:10.2527/jas.2014-7836
Kim, S. W., Jhon, M., Kim, J. M., Smesny, S., Rice, S., Berk, M., . . . Amminger, G. P. (2016). Relationship between Erythrocyte Fatty Acid Composition and Psychopathology in the Vienna Omega-3 Study. PLoS One, 11(3), e0151417. doi:10.1371/journal.pone.0151417
Paduchova, Z., Katrencikova, B., Vavakova, M., Laubertova, L., Nagyova, Z., Garaiova, I., . . . Trebaticka, J. (2021). The Effect of Omega-3 Fatty Acids on Thromboxane, Brain-Derived Neurotrophic Factor, Homocysteine, and Vitamin D in Depressive Children and Adolescents: Randomized Controlled Trial. Nutrients, 13(4). doi:10.3390/nu13041095
Papandreou, P., Gioxari, A., Ntountaniotis, D., Korda, O. N., Skouroliakou, M., & Siahanidou, T. (2020). Administration of an Intravenous Fat Emulsion Enriched with Medium-Chain Triglyceride/omega-3 Fatty Acids is Beneficial Towards Anti-Inflammatory Related Fatty Acid Profile in Preterm Neonates: A Randomized, Double-Blind Clinical Trial. Nutrients, 12(11). doi:10.3390/nu12113526
Trebaticka, J., Hradecna, Z., Surovcova, A., Katrencikova, B., Gushina, I., Waczulikova, I., . . . Durackova, Z. (2020). Omega-3 fatty-acids modulate symptoms of depressive disorder, serum levels of omega-3 fatty acids and omega-6/omega-3 ratio in children. A randomized, double-blind and controlled trial. Psychiatry Res, 287, 112911. doi:10.1016/j.psychres.2020.112911
7. Do these omega-3 FAs affect beneficially the glymphatic system, i.e. waste clearance pathway in the brain, dedicated to drain away soluble waste proteins and metabolic products in case of Parkinson and Alzheimer patients? What is the added value to measure EPA/DHA in cerebrospinal fluid?
Thanks for your comments. Although we would love to include all aspects concerning omega 3, this paper focuses mainly on CVD. We have selected a handful of references and information to support the critical information. Although pertinent, we consider your question a better fit for papers dedicated to omega-three and CNS.
Reviewer 2 Report
The authors propose an interesting review of the current literature on omega 3, there are some points to review:
- Even if it is a narrative review, how were the articles selected?
- The antiinflammatory effect in the prostaglandin pathway and resolvins, in particular, should at least be mentioned
- In general, the primary mechanism should be proposed, possibly with the help of a figure
- In the dosage paragraph, the sources and extraction/concentration methods that can be a source of non-uniform results should also be emphasized
- In the conclusions, it would be helpful to indicate possible dosages of use, both as prevention and as treatment; as well as suggesting possible studies to operate
Author Response
- Even if it is a narrative review, how were the articles selected? Multiple articles were selected using PubMed to build the most updated evidence-based paper on omega 3 FA. Supporting papers to explain further the details pertinent to omega 3 FA were also selected based on the year of publication and journals (looking for more recent data). We independently screened all titles and abstracts of the retrieved studies. Disagreements regarding the inclusion of the studies and the interpretation of the data were resolved by discussion among investigators.
- The anti-inflammatory effect in the prostaglandin pathway and resolvins, in particular, should at least be mentioned. Thanks for your comments. Please refer to lines:2429 "Oxidative stress is a significant cause of heart fibrosis; it is highly implicated in the development of HF and is partly controlled by the nuclear factor erythropoietin two related factor 2 (NRF2). EPA, DHA, and specialized pro-resolving lipid mediators (SPMs), such as resolvin D1 (RvD1), can activate NRF2, which could protect the heart from the onset of cardiac fibrosis"2998 "Ω3FA has been demonstrated to play a role in this cytokine storm by being a substrate of plasma and synovial fluid specialized pro-resolving mediators (SPMs) such as the recombinant bivalent fusion protein (RvEs), protectin D1 or neuroprotectin D1 (PD1), protectin DX (PDX), and the pro-resolving lipid mediator maresin 1 (MaR1), which reduce circulating inflammatory cytokines, thereby reducing inflammatory-mediated diseases such as rheumatoid arthritis (RA) disease activity [88,89]. This has been later demonstrated by an inverse correlation between the O3I and disease activity [90]."
-
In general, the primary mechanism should be proposed, possibly with the help of a figure. Thanks for your suggestion, the mechanism is specified in the text.
- In the dosage paragraph, the sources and extraction/concentration methods that can be a source of non-uniform results should also be emphasized. Thanks for your comments. This is described under the index section.
- In the conclusions, it would be helpful to indicate possible dosages of use, both as prevention and as treatment; as well as suggesting possible studies to operate. Thanks for your comments, the dose, as emphasized by our metanalysis and by the REDUCE IT trial has been clarified and added to the dose paragraph and the conclusions.
Round 2
Reviewer 2 Report
I think the authors have improved the manuscript, but I still have to report:
- about the concentration, my note concerns the type and concentration of omega3 in the supplements not in the diet, in the index section, in the same way in the dosages section: for example, in the conclusions, we indicate two doses of 2g per day, so 4g but of DHA + EPA or 4g of a supplement that contains DHA and EPA, because the two are completely different!
